# EdgeCaseDNet: An enhanced detection architecture for edge case perception in autonomous driving

**Yin Lei** ◐, **Chen Shan**◐, **Wancheng Ge**\*

College of Electronic and Information Engineering, Tongji University, Shanghai, China

◐ These authors contributed equally to this work.
\* gwc828@tongji.edu.cn

## Abstract

Autonomous driving perception systems still encounter significant challenges in edge scenarios involving multi-scale target changes and adverse weather, which seriously compromise detection reliability. To address this issue, we introduce a novel edge case dataset that extends existing benchmarks by capturing extreme road conditions (fog, rain, snow, nighttime et al.) with precise annotations, and develop EdgeCaseDNet as an optimized object-detection framework. EdgeCaseDNet's architecture extends YOLOv8 through four synergistic innovations: (1) a Haar_HGNetv2 backbone that enables hierarchical feature extraction with enhanced long-range dependencies, (2) an asymptotic feature pyramid network for context-aware multi-scale fusion, (3) a hybrid partial depth-wise separable convolution module, and (4) Wise-IoU loss optimization for accelerated convergence. Comprehensive evaluations demonstrated the superiority of EdgeCaseDNet over YOLOv8, achieving improvements of +10.6% in mAP@50, and +8.4% in mAP@[.5:.95]. All the relevant codes are available at https://github.com/yutianku/EdgeCaseDNet.

## 1. Introduction

The autonomous driving industry has developed rapidly, revolutionizing transportation and safety. Compared with traditional driving systems, autonomous vehicles have the potential to significantly reduce accidents, enhance traffic efficiency, and improve the overall mobility. However, the rapid development of this industry is significantly challenged by the accurate detection of road targets, which is crucial for safe navigation and decision making. Numerous studies have addressed the problem of road target detection. Methods such as YOLOv3 [1], YOLOv4 [2], YOLOv8 [3], RT-DETR [4],and YOLOv12 [5] have been proposed and developed to provide valuable tools for object detection in road scenes. While existing methods demonstrate satisfactory multi-target recognition performance on general-purpose datasets such as COCO, they still face significant limitations in addressing edge cases encountered in real-world road

**Data availability statement:** All the relevant codes are available at https://github.com/yutianku/EdgeCaseDNet.

**Funding:** This research was supported in part by the International Strategic Innovative Project of the National Key Research and Development Program of China under Grant (No. 2023YFE0112500). The funders contributed to the research by providing the research topic and offering financial support which was used to purchase deep learning training servers essential for our study. Furthermore, they played a supportive role in the decision to publish and preparation of the manuscript.

**Competing interests:** The authors have declared that no competing interests exist.

scenarios, particularly lacking specialized datasets capturing nuanced traffic situations (e.g., extreme illumination changes and non-standard obstacles) and corresponding tailored detection methodologies.

In real-world road scenarios, object detection models often encounter edge cases, where detection becomes exceptionally challenging owing to factors such as severe occlusion, adverse weather conditions, low illumination, small object size, or complex background interference. These cases deviate from the typical distribution of training data and can lead to high rates of false negatives and positives, posing serious safety risks to autonomous driving systems. To address this critical issue, we propose a new dataset for edge cases in autonomous driving perception (DEAP), which is specifically designed to evaluate the model performance under demanding conditions. This study addresses the crucial problem of enhancing the accuracy and reliability of object detection in road scenes, with a specific focus on achieving robust edge-case perception in autonomous driving under adverse conditions.

This study makes two major contributions to address these limitations.

(1) The DEAP dataset is a new benchmark specifically designed to evaluate object detection models under extreme road conditions. Built upon data from SODA10M [6], CODA [7] and DAWN [8] and enriched through advanced augmentation techniques, DEAP provides a realistic and comprehensive testbed for assessing the robustness and generalization of vision-based perception systems.

(2) An improved detection architecture called EdgeCaseDNet is built on the YOLOv8 framework. EdgeCaseDNet incorporates several components: a Haar wavelet-based HGNetv2(Haar_HGNetv2) backbone inspired by PP-HGNetv2 [9] for better feature representation, an asymptotic feature pyramid network(AFPN) module for enhanced multiscale fusion, a lightweight hybrid partial depthwise separable convolutional module (HPDSConv) to reduce the computational load, and the Wise-IoU [10] loss function to improve regression accuracy and convergence behavior.

Together, these innovations enable EdgeCaseDNet to deliver higher accuracy and robustness under challenging real-world conditions while maintaining an efficient inference speed. The primary contribution of this study lies in the domain-adaptive integration and joint optimization of these components to effectively address the characteristic challenges in autonomous driving edge cases, such as the presence of rare objects, ambiguous scene backgrounds and low signal-to-noise ratios.

The remainder of this paper is organized as follows. Section 2 reviews the literature on the small-object detection of road targets. Section 3 details the enhancements made to the YOLO model, while Section 4 presents the outcomes of comparative and ablation studies along with discussions. Section 5 concludes the paper with a summary of the findings.

## 2. Related work

Object detection algorithms are crucial for environmental perception in computer vision for autonomous driving. However, owing to the immense scale variation

of targets, their susceptibility to occlusion, and the complex and dynamic nature of road backgrounds, traditional object detection algorithms often struggle to achieve high accuracy. Consequently, it is imperative to devise an object detection framework tailored to the intricacies of complex road environments. Traditional road object detection methods primarily rely on manually extracting image features using techniques such as Haar [11], SIFT [12], and LBP [13] and then applying classifiers such as SVM [14] and AdaBoost [15]to these manually extracted elements. These traditional object detection algorithms rely on handcrafted features [16], and they may struggle with scale variations, occlusions, and dynamic lighting conditions in complex traffic scenarios. Nonetheless, these approaches have established a foundational framework to advance subsequent detection methodologies [17].

Deep-learning-based object detection methods have been extensively adopted in road scenario. Vehicle-target recognition under complex weather conditions has been addressed using a domain-adaptive DAGL-Faster algorithm [18]. This algorithm improves recognition accuracy by enhancing the domain feature extraction capability through the integration of domain classifiers. A transformer-based multi-scale feature fusion pedestrian detection network that utilizes gating mechanisms and feature enhancement to repress irrelevant features and read global image information via a transformer network, effectively addressing long-distance dependability issues [19].

The YOLO algorithm is renowned for its speed and accuracy in object detection. Researchers have enhanced its performance to adapt more effectively to diverse scenarios in recent years. Their efforts primarily concentrated on refining the feature integration capabilities of the algorithm, thereby improving its object recognition proficiency. The YOLO algorithm was enhanced by substituting the original modules with the C2f_RFAConv module, which improved feature extraction efficiency [20]. The integration of the Triplet Attention mechanism enhances feature concentration, thereby significantly boosting the overall efficacy of the YOLOv8 framework. The YOLOv8 algorithm was improved by integrating the Coordinate Attention (CA) module to enhance feature extraction and object localization in complex scenarios, addressing limitations of conventional detection methods [21]. A YOLOv8-based network (YOLOv8-qsd) was developed using structural reparameterization techniques to transform models built upon diverse branch blocks (DBB) [22]. To accurately detect small targets, the algorithm integrates features of varying scales and implements a feature pyramid based on a Bidirectional Feature Pyramid Network (BiFPN) [23] after the backbone. A novel pipeline structure model based on queries is introduced to address the challenge of long-range detection in driving scenarios. The test results demonstrate that this algorithm outperforms YOLOv8 in terms of both speed and accuracy on a large-scale small-target detection dataset (SODA-A), maintaining efficiency while ensuring detection accuracy. The integration of attention mechanisms into the YOLOX model improved detection accuracy without compromising processing speed, as validated on multiple datasets [24]. The YOLOv5 algorithm was optimized by integrating the SE attention mechanism and replacing the feature extraction backbone with MobileNetV3, leading to significant improvements in both accuracy and inference speed [25]. The YOLOv5 algorithm was adapted for drone-based object detection by incorporating a coordinate attention mechanism, which notably improves detection accuracy for small targets [26]. An improved BiFPN was incorporated into the YOLOv8 network to enhance feature fusion capabilities and address challenges such as high computational cost and suboptimal detection precision [27]. Despite the promising results of these studies on single-target detection under specific road scenarios, there remains room for performance improvements in detecting multiple targets in complex backgrounds and road environments of varying scales. To address the issue of traditional downsampling operations that cause the loss of important spatial information in semantic segmentation tasks with complex backgrounds, the haar wavelet-based downsampling (HWD) module was introduced to enhance the performance of semantic segmentation models [28].

YOLOv8 is a highly popular technical framework for identifying objects and is an iterative enhancement of the YOLO series of algorithms. Since its introduction, the YOLO series has garnered significant attention because of its

exceptional performance in real-time object detection. To address the inherent challenges in real-world road scenarios, such as diverse target types, varying sizes, and complex background interference, and to mitigate issues such as poor detection precision and high false-negative rates, this study presents enhancements to the YOLOv8 algorithm.

## 3. Methods

### 3.1. The structure of EdgeCaseDNet

YOLOv8 demonstrates exceptional performance in detecting targets of conventional scales, but exhibits certain limitations in handling diverse target types, scales, and complex backgrounds in road scenarios. The range of target types in road environments, including pedestrians, vehicles, and traffic signs, varies in size and is influenced by multiple complex factors such as lighting changes and dynamic backgrounds, which can lead to false negatives and false positives. To address the deficiencies of the original YOLOv8 algorithm in edge scenarios, targeted modifications were made to the YOLOv8's backbone, neck, and head components in this study. Fig 1 illustrates the EdgeCaseDNet network architecture based on an improved YOLOv8 framework.

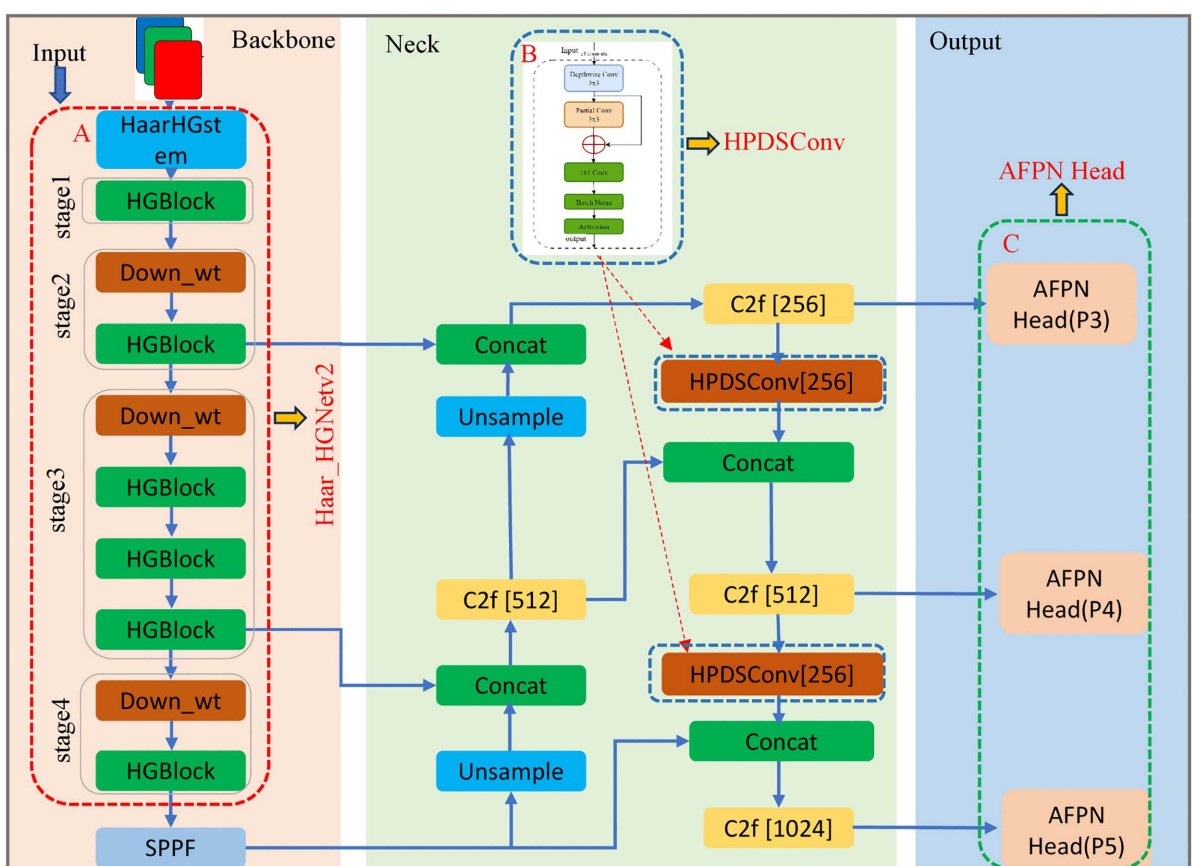

**Fig 1. The architectural framework of EdgeCaseDNet.** The red dashed lines box shows the optimized elements of the improved network structure. The red dashed lines in box A show the backbone replaced by the Haar_HGNetv2 module. In box B, the upsampling module is replaced by the HPDSConv module. In box C, the head network with AFPN added. For a detailed examination of the HPDSConv module, please refer to Fig 3.

## 3.2. The motivation and design of the Haar_HGNetv2 backbone

Conventional downsampling operations (e.g., strided convolution or pooling) in CNN-based backbones often suffer from information loss during spatial compression, particularly under multi-scale target variations and adverse weather conditions. This limitation arises from their local receptive fields and sensitivity to high-frequency noise, which hinders the preservation of critical edge features in extreme scenarios. To address this, we propose Haar wavelet-based downsampling (Down_wt) module that replaces the standard downsampling layers in HGNetv2. Unlike heuristic pooling strategies, wavelet decomposition provides a mathematically rigorous multi-resolution analysis framework. Specifically, Haar wavelet transforms project input features into orthogonal subspaces (approximation and detail coefficients) through linear operations, enabling the explicit separation of low-frequency global structures and high-frequency local details. This property aligns with the requirement of autonomous driving perception systems to maintain robustness against scale shifts and weather-induced artifacts.

Haar wavelets are particularly advantageous for edge case perception because of their inherent structural and computational characteristics. Their piecewise-constant basis exhibits high sensitivity to abrupt intensity changes, such as those observed in pedestrians emerging from shadows or roadside debris, rendering them more effective than smoother wavelets (e.g., Daubechies wavelets) in capturing sharp spatial transitions under degraded conditions. Furthermore, the Haar transform requires only addition and subtraction operations, facilitating ultralow-latency, parameter-free multiscale decomposition. The resulting subbands (LL, LH, HL, and HH) provide explicit representations of both the coarse structure and directional high-frequency details, enabling the selective enhancement of critical features (e.g., HH for small debris) without increasing model complexity. This offers a lightweight yet robust alternative to deeper CNNs, which often over-smooth rare anomalies.

Let $(X \in R^{C \times H \times W})$ denote an input feature map. The 2D Haar wavelet transform was applied to separable filtering along spatial dimensions using low-pass $\left( (h_{low} = \left[ 1/\sqrt{2}, 1/\sqrt{2} \right] ) \right)$ and high-pass $\left( (h_{high} = \left[ 1/\sqrt{2}, -1/\sqrt{2} \right] ) \right)$ filters. For each channel, the transformation yields four sub bands which can be expressed by Eq (1).

$$
\begin{cases}
LL = h_{low} * (h_{low} * X)^{\top} \\
LH = h_{high} * (h_{low} * X)^{\top} \\
HL = h_{low} * (h_{high} * X)^{\top} \\
HH = h_{high} * (h_{high} * X)^{\top}
\end{cases}
\tag{1}
$$

Where $*$ denotes a 1D convolution applied row-wise or column-wise, and each output is implicitly downsampled by a factor of 2 (i.e., retaining every other sample). The LL subband captures coarse low-frequency approximations, whereas LH, HL, and HH encode horizontal, vertical, and diagonal detail coefficients, respectively.

This design confers three principal advantages: (1) aliasing suppression: the orthogonal Haar basis ensures strict frequency separation, thereby minimizing spectral leakage; (2) multiscale feature preservation: explicit decomposition retains both global structure and local anomalies; and (3) parameter efficiency: the transform employs fixed, non-learnable kernels, thereby eliminating trainable parameters and reducing FLOPs compared to standard convolutions, while remaining fully differentiable and compatible with end-to-end training.

By integrating Down_wt into HGNetv2 to construct Haar_HGNetv2, the backbone gains an enhanced capacity to propagate both coarse- and fine-grained edge information through the network hierarchy, which is particularly beneficial for detecting distorted objects in adverse conditions. HGNet consists of multiple HG blocks, the details of which are shown in Fig 2.

The number of regular convolutions increases with the depth of the layers, resulting in a backbone network that is advantageous for GPU inference. In addition, HGNet has the following features.

1) The first layer of HGNet is composed of a stem module with 96 channels, which helps reduce the number of parameters and computational load to a certain degree.

2) The Tiny architecture of HGNet is built by four HG Stages, each of which mainly consists of HG Blocks containing numerous standard convolutions.

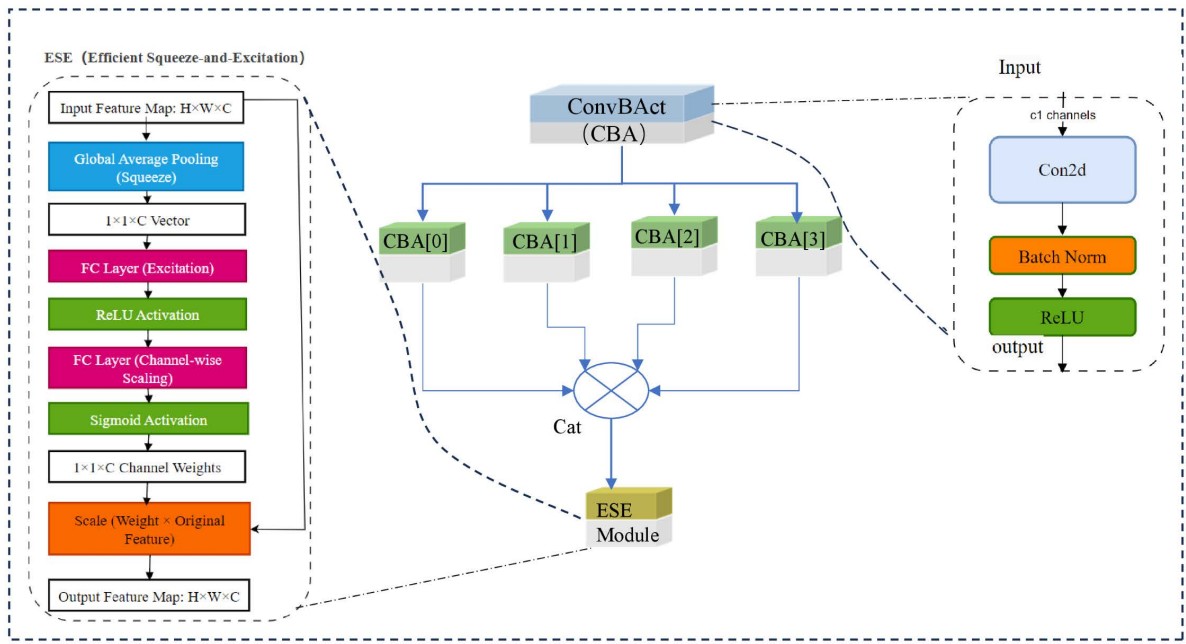

**Fig 2. The network architecture of HG-Block.** In the HG-Block architecture, the ESE module is integrated into the network structure to enhance feature extraction and representation capabilities. The convolution batch Norm activation (ConvBAct module) is composed of a convolutional layer (Conv), batch normalization layer (BatchNorm), and activation function (Activation). It serves as a fundamental building block for feature extraction and feature transformation in deep neural networks.

3) The third to fifth layers of HGNet use learnable downsampling layer modules (LDS Layers), where the number of groups matches the input channels, further reducing model parameters and computational requirements. The Tiny model employs three LDS layers. This design choice strikes an optimal balance between reducing the computational load and maintaining effective feature extraction, thereby maximizing the GPU computation efficiency.

4) HGNet employs the ReLU activation function, with constant-time operations ensuring that the model maintains its fast inference speed on hardware.

### 3.3. HPDSConv

The HPDSConv module constitutes a lightweight convolution operation that combines depth-wise separable convolution (DSConv) [29], which is used to reduce the number of parameters and computational complexity of the model, and Partial Convolution (PConv), which is used to address incomplete or missing data. HPDSConv can be used to process images with complex backgrounds and multi-scale targets. The structure of HPDSConv is illustrated in Fig 3.

### 3.4. Asymptotic feature pyramid network

Road environment detection is a challenging task for autonomous driving systems, particularly in terms of detecting objects at various scales and real-time detection. During the object detection process, a significant variation in object size affects the detection accuracy. Therefore, this study introduces a feature pyramid network, replacing the original feature pyramid network in YOLOv8 with the AFPN [30] to enhance the detection performance of YOLOv8 on multi-scale targets while maintaining real-time capabilities. The structure of the AFPN is shown in Fig 4.

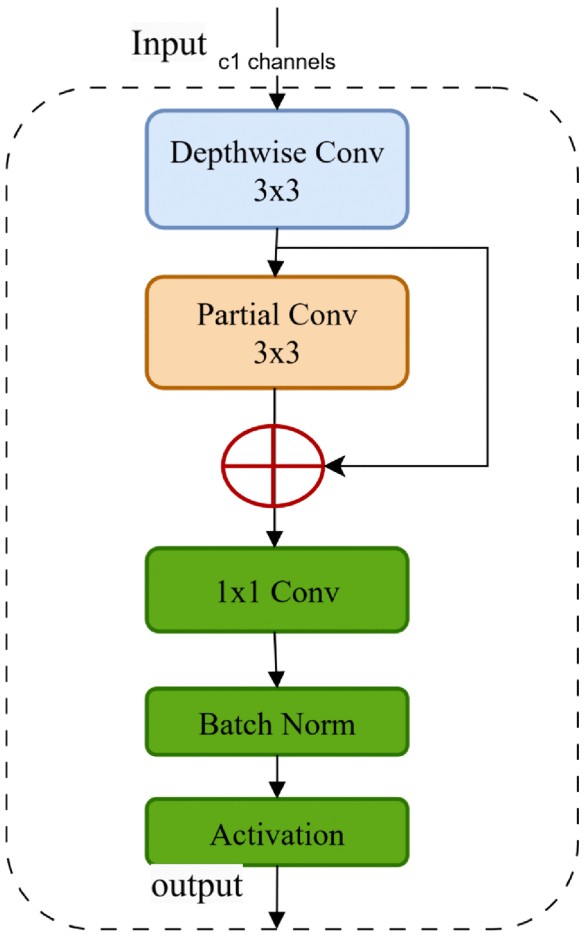

**Fig 3. Structure of the HPDSConv.**

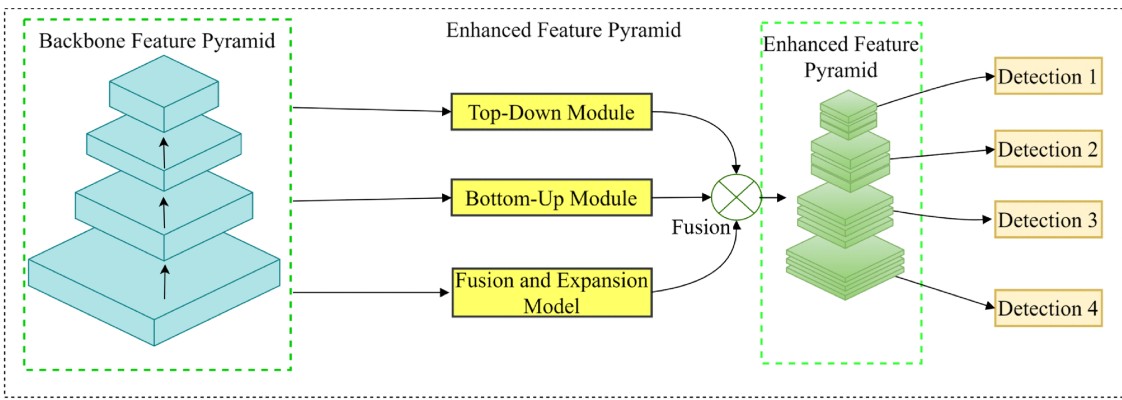

**Fig 4. Structure of the AFPN.**

## 3.5. Loss function

To enhance the conventional Intersection over Union (IoU) metric, the original boundary regression loss function in YOLOv8 is called CIoU. CIoU considers the ratio, shape, size, and overlapping area inside the bounding boxes, in addition to the distance between their centers. It is primarily used in computer vision fields such as video surveillance. The ground truth boxes are shown by the green rectangles in Fig 5, whereas the anticipated boxes are represented by red rectangles.

The following is the definition of the CIoU loss function formula (Eqs (2)–(4)):

$$L_{CIoU} = 1 - IoU + \frac{\rho^2(b_g, b_p)}{c^2} + \alpha V \tag{2}$$

$$V = \frac{4}{\pi^2} \left( \arctan \frac{w_g}{h_g} - \arctan \frac{w_p}{h_p} \right)^2 \tag{3}$$

$$\alpha = \frac{V}{1 - IoU + V} \tag{4}$$

In this case, the central points of the ground truth and predicted boxes are denoted by bg and bp, respectively, and the diagonal length of the smallest rectangle containing both predicted and ground truth boxes can be expressed as c. The Euclidean distance between the centers of the ground truth and predicted boxes is embodied by $\rho$, where the width and height of the ground truth and predicted boxes are denoted by Wg and Hg, respectively, and the ratio of the intersection and union of the ground realness and predicted boxes is represented by the IoU. It is evident from the above that V in the CIoU approaches 0 and loses its penalizing effect when the width and height of the ground truth box and the forecast box are equal, making it challenging to optimize the loss. Furthermore, the predicted box height hg and width wp cannot be changed simultaneously, which prevents the algorithm from improving.

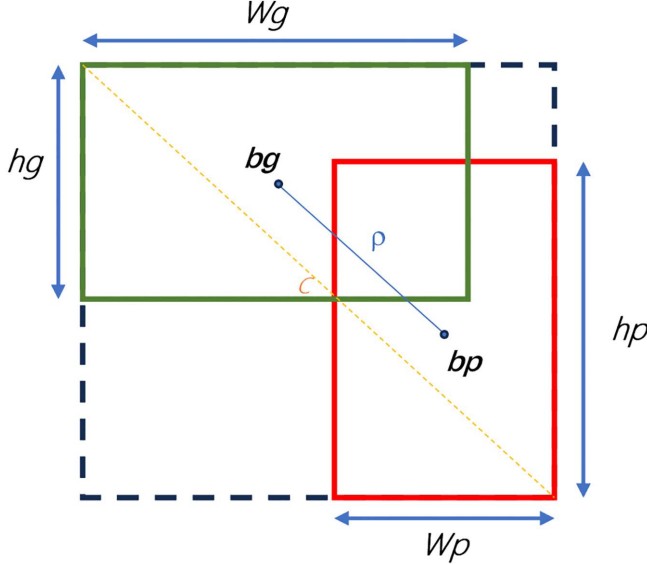

**Fig 5. CIoU.**

To address these issues, we introduced a novel bounding box regression loss function called Wise-IoU, which is inspired by the geometric properties of axis-aligned rectangles. Reducing the distances between the top-left and bottom-right corners of the anticipated and ground-truth boxes accelerates the algorithm's convergence and increases its localization precision. The loss function uses a dual-attention methodology that boosts the accuracy and effectiveness of the object detection. The parameter $R_{WIoU} \in [1, e]$ is crucial for adjusting the model's emphasis on anchors with moderate confidence scores, which are referred to as anchors of average quality. These anchors typically represent regions of interest that are neither highly certain nor extremely uncertain, thus balancing the precision and recall during training. When $L_{IoU} \in [0, 1]$, the focus shifts away from high-quality anchors. Instead, when anchors are well-aligned with the target boxes, particular emphasis is placed on the separation between their centers. The relationship between $R_{WIoU}$ and $L_{IoU}$ can be expressed as Eq (5).

$$L_{WIoUv1} = R_{WIoU} L_{IoU} \tag{5}$$

The value of $R_{WIoU}$ is determined by Eq (6):

$$R_{WIoU} = \exp\left(\frac{(x - x_{gt})^2 + (y - y_{gt})^2}{\left(W_g^2 + H_g^2\right)_*}\right) \tag{6}$$

The smallest width and height of the enclosing bounding box were represented by the values of Wg and Hg. To guarantee that the optimization process proceeds without hiccups and to avoid $R_{WIoU}$ convergence problems during gradient updates, Wg and Hg were decoupled from the computation graph (indicated by the superscript). This approach effectively removes factors that could impede model convergence and avoids introducing additional complexity metrics such as the aspect ratio.

## 4. Results

### 4.1. Dataset & experimental environment

The development and benchmarking of autonomous driving perception systems are highly dependent on high-quality, annotated datasets. However, current publicly available datasets generally suffer from limitations in scene coverage, with over 98% of the annotated data collected under well-lit, clear weather conditions, and typical driving scenarios. This limitation directly leads to safety hazards, such as target misses and misclassifications in adverse weather conditions (e.g., fog, rain, snow, and nighttime) and other edge scenes, such as abnormal road conditions. To address these issues, we introduced the DEAP dataset, which focuses on diverse weather scenarios for edge cases in autonomous driving. The core innovations of DEAP are reflected in three main aspects. First, the dataset includes four typical adverse weather conditions and extreme driving scenarios, integrating over 2,000 edge case scenarios through a multi-source data fusion strategy. Some samples were selected from the authoritative open-source datasets, CODA and SODA10M, and subjected to rigorous quality screening and scenario validation. All samples have been manually verified to exhibit at least one of the following characteristics: (i) extremely rare object categories, (ii) severe occlusion, motion blur, or extreme scale variation, or (iii) adverse environmental conditions (e.g., heavy rain, fog, snow, low illumination) that lead to low signal-to-noise ratios in visual perception.The DEAP dataset features a structured data system comprising 14,057 annotated samples. The dataset was partitioned into training (9,837 samples), validation (2,809 samples), and testing (1,411 samples) sets following an approximate 7:2:1 ratio, using a two-stage stratified splitting strategy to ensure both representativeness and generalization. Specifically, 15% of the samples from each of the five edge-case condition groups(night, rain, fog, snow, and abnormal traffic) were first reserved for the test set; the remaining data were then split into training and validation sets via joint stratification over the object category and condition label, guaranteeing that all classes appear across all splits and environmental contexts. To further enrich the visual diversity, we applied data augmentation techniques, including random rotation, scaling, and partial occlusion, during training. All images were annotated using the semi-automatic labeling

tool X-AnyLabeling [31], with every sample manually verified to exhibit at least one defining characteristic of edge cases: semantic rarity, severe occlusion/motion blur, or adverse environmental conditions that led to low signal-to-noise ratios. Full per-class, per-condition, and per-split statistics are provided in Tables 1 and 2 and will be released alongside the dataset to ensure full reproducibility.

The long-tailed nature of DEAP reflects real-world deployment challenges but may disadvantage standard training pipelines. The future work could explore class-balanced losses (e.g., focal loss), copy-paste augmentation, or few-shot adaptation to better handle tail classes.

In this experiment, we used a Windows operating system with an NVIDIA GeForce RTX 4090 GPU. The deep learning architecture used was PyTorch 2.0, with CUDA version 11.8, and the chosen programming language was Python 3.9. A total of 200 training epochs were chosen. To ensure fair comparisons, all comparison algorithms and ablation studies were conducted under the same experimental conditions and training parameters.

## 4.2. Evaluation metrics

The criteria used for evaluation included precision (P), recall (R), mean Average Precision (mAP), and mAP@[.5:.95]. A higher P indicates greater accuracy of the detection results, meaning fewer false positives. A higher R signifies that the algorithm can detect as many targets as possible, meaning fewer false negatives. An increased mAP value signifies increased algorithmic detection precision. The following formulae (Eqs (7)-(10)) can be used to determine these parameters.

$$P = \frac{TP}{TP + FP}$$

(7)

$$R = \frac{TP}{TP + FN}$$

(8)

**Table 1. Per-class train/val/test instance counts in DEAP dataset.**

| Object Class | Train | Val | Test | Total |
|---|---|---|---|---|
| Image Amount | 9837 | 2809 | 1411 | 14057 |
| Car | 42591 | 11905 | 6114 | 60610 |
| Truck | 10828 | 3097 | 1372 | 15297 |
| Bus | 2493 | 728 | 350 | 3571 |
| Bicyclist | 7377 | 2034 | 1085 | 10496 |
| Pedestrian | 7355 | 2053 | 1036 | 10444 |
| Motorcycle | 349 | 71 | 41 | 461 |
| Total | 70993 | 19888 | 9998 | 100879 |

**Table 2. Cross-tabulation of object instances by edge condition and class.**

| Condition/ Class | Car | Truck | Bus | Bicyclist | Pedestrian | Motorcycle | Total per condition | Image Amount |
|---|---|---|---|---|---|---|---|---|
| Night | 954 | 81 | 26 | 173 | 221 | 7 | 1462 | 492 |
| Rain | 2528 | 238 | 34 | 45 | 120 | 5 | 2970 | 657 |
| Fog | 5126 | 608 | 166 | 112 | 426 | 82 | 6520 | 1207 |
| Snow | 3081 | 196 | 53 | 85 | 395 | 1 | 3811 | 644 |
| Abnormal Traffic | 48921 | 14174 | 3292 | 10081 | 9282 | 366 | 86116 | 11057 |
| Total | 60610 | 15297 | 3571 | 10496 | 10444 | 461 | 100879 | 14057 |

$$AP = \int_0^1 P(R)dR \qquad (9)$$

$$mAP = \frac{1}{N}\sum_{i=1}^{N} AP_i \qquad (10)$$

The model performance was evaluated using standard object detection metrics, specifically true positives (TP), false positives (FP), and false negatives (FN), which were determined based on an Intersection over Union (IoU) threshold of 0.5 between the predicted and ground-truth bounding boxes. The Average Precision (AP) was computed as the area under the precision-recall curve for each class, and the mean Average Precision (mAP) was reported as the average AP across all classes.

### 4.3. Comparison with state-of-the-art

This study undertakes a comparative evaluation of EdgeCaseDNet in relation to recent open-vocabulary and real-time detection models under strictly controlled experimental conditions. In particular, YOLO-World is examined in a zero-shot framework using a consistent set of 10 corner-case text prompts, such as "pedestrian walking on urban sidewalk"and"cyclist riding bicycle on road."YOLOv8 and RT-DETR were implemented using their official weights and default settings. All models, including ROAD's Faster R-CNN and SSD, were assessed on the DEAP dataset without any fine-tuning, utilizing uniform pre- and post-processing procedures. To ensure a rigorous and reproducible evaluation, we conducted five independent trials, with all improvements over baseline measures being statistically significant ($p < 0.01$, paired t-test).

The findings, detailed in Table 3, reveal that EdgeCaseDNet achieves the highest mAP@[.5:.95] at 37.2% and recall@100 at 51.9%, underscoring its superior capability in detecting edge cases compared to all currently reproducible baselines.

To evaluate cross-domain robustness, a zero-shot assessment was conducted on a curated subset of the BDD100K dataset, comprising 1,011 images depicting adverse conditions selected based on official weather condition annotations. The EdgeCaseDNet model, which was exclusively trained on DEAP, was assessed without fine-tuning. Predictions were confined to the six object categories common to both datasets(car, truck, bus, pedestrian, bicyclist, and motorcycle). Under this protocol, our method achieved a mean Average Precision (mAP) of 43.4%, surpassing the YOLOv8n baseline (40.0% mAP) by 3.4%, thereby demonstrating enhanced generalization to previously unseen adverse driving scenarios.

The inference efficiency was assessed on a mobile laptop (Intel i7-10510U CPU, NVIDIA GeForce MX250), resulting in a latency of 27.9 ms and 35.8 FPS under full load. Detailed results, including per-component breakdown and comparisons with baseline methods, are provided in Table 4.

**Table 3. Comparison of the performance of different algorithms.**

| Algorithm | $AP_{50}^{val}$ | $AP_{50:95}^{val}$ | P% | R% |
|---|---|---|---|---|
| SSD | 38.13 | / | 84.4 | 69.2 |
| RT-DETR | 35.9 | 22.7 | 63 | 34 |
| Yolov6n | 43.9 | 27.4 | 63.2 | 40.4 |
| Yolov8n | 47.7 | 30.7 | 70.1 | 43.7 |
| YOLO11n | 46.5 | 29.7 | 59.9 | 41.4 |
| YOLOv12n | 34.6 | 21 | 51.2 | 31.5 |
| YOLO-World | 38.2 | 22.5 | 68.3 | 42.8 |
| EdgeCaseDNet(ours) | **58.3** | **39.1** | **68.4** | **50.4** |

**Table 4. Assessment of the model's inference efficiency on mobile devices.**

| Metric | EdgeCaseDNet | YOLOv8n (baseline) |
|---|---|---|
| GFLOPs | 7.0 | 8.2 |
| Parameters | 2.5M | 3.0M |
| Input size | 640x640 | 640x640 |
| latency | 27.9 ms | 54.4ms |
| FPS | 35.8 | 18.3 |
| Peak memory usage | 417MB | 432MB |

### 4.4. Object detection experiments in edge cases

Additionally, DEAP now includes five safety-critical edge-case categories: (i) adverse weather conditions (rain/fog), (ii) low-light/night scenarios, (iii) visual interference (e.g.,ground reflection), (iv) severe visual degradation (heavy snow/sandstorm), and (v) anomalous traffic events (e.g., overturned vehicles). This expanded scope ensures comprehensive coverage of the challenging conditions pertinent to real-world autonomous driving deployments. The test results for the five scenarios are presented in Figs 6–10. The inference results indicate that mainstream

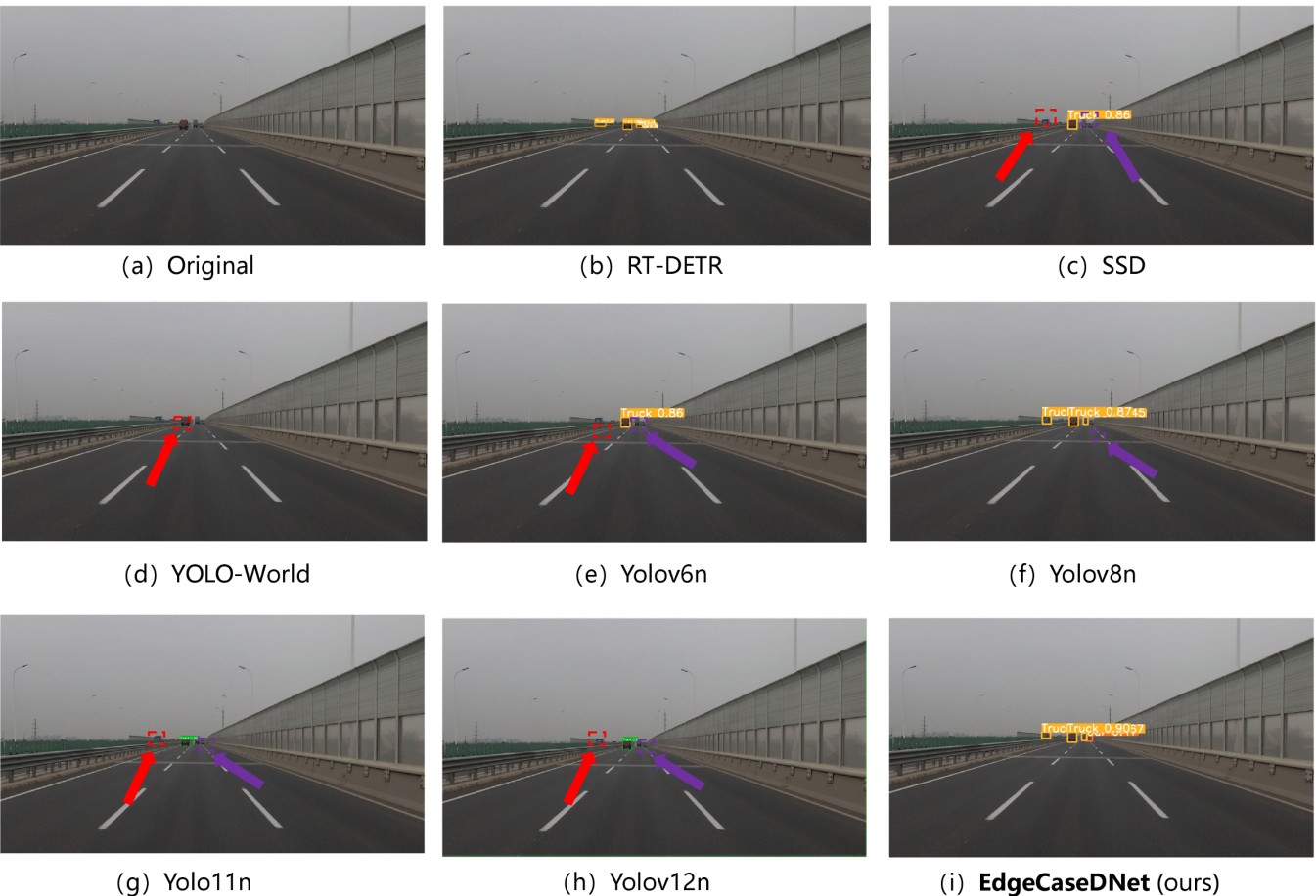

**Fig 6. Comparison of the target detection results of different models in Scenario 1.** Red arrows indicate missed detections due to partial occlusion of the targets, while purple arrows highlight missed detections of small, blurry objects. Additionally, the detection results shown in image (a) reveal that the RT-DETR model, although not missing any targets, exhibited multiple detections of the same target with low confidence scores.

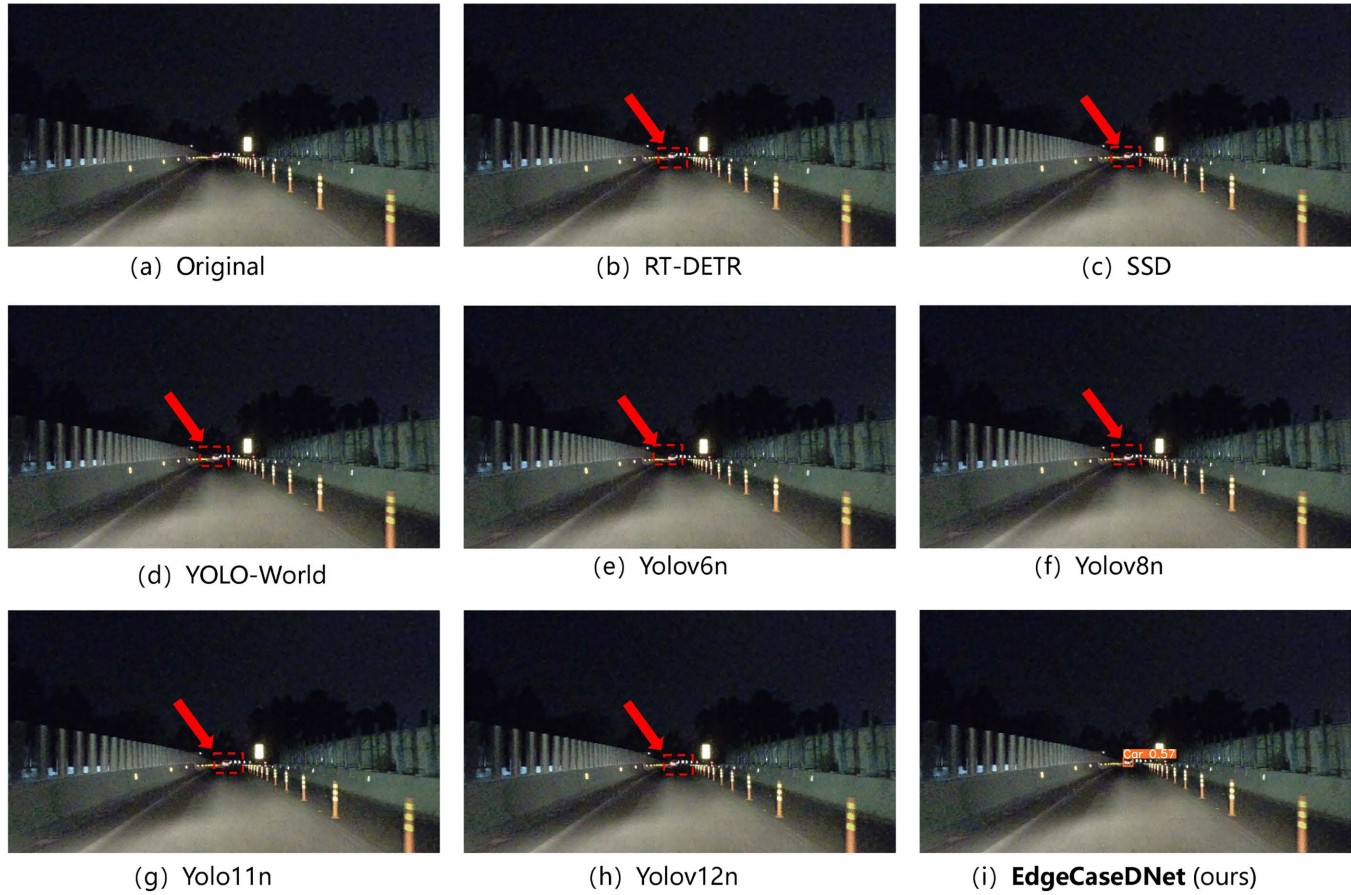

**Fig 7. The detection results of different algorithms in Scenario 2.** Red arrows indicate missed detections due to insufficient nighttime illumination and a blurry background. The detection results in images (b)–(h) demonstrate that conventional object detection algorithms fail to effectively detect targets in low-light, edge-case scenarios with blurry backgrounds. In contrast, only the optimized network framework algorithm, EdgeCaseDNet, successfully detected the targets.

conventional object detection methods exhibit varying degrees of missed and false detections in these edge case scenarios. In contrast, the improved detection framework EdgeCaseDNet enhances the ability of the algorithm to detect targets of different scales, thereby demonstrating its effectiveness and superiority in edge-case road target detection.

Although EdgeCaseDNet demonstrated robust performance in most edge cases, it failed to detect an overturned vehicle in one test sequence (see Fig 10). Further analysis indicated that this failure was primarily attributable to data scarcity; existing training datasets, including DEAP, contained virtually no instances of vehicles in fully inverted positions. Consequently, the model did not acquire the ability to recognize discriminative features for this configuration and tended to misclassify it as background or non-vehicle clutter. This underscores a fundamental limitation of data-driven methodologies when faced with ultra-rare events that are absent from the training distribution, thereby suggesting the necessity of synthetic data augmentation or simulation-based training to address such scenarios.

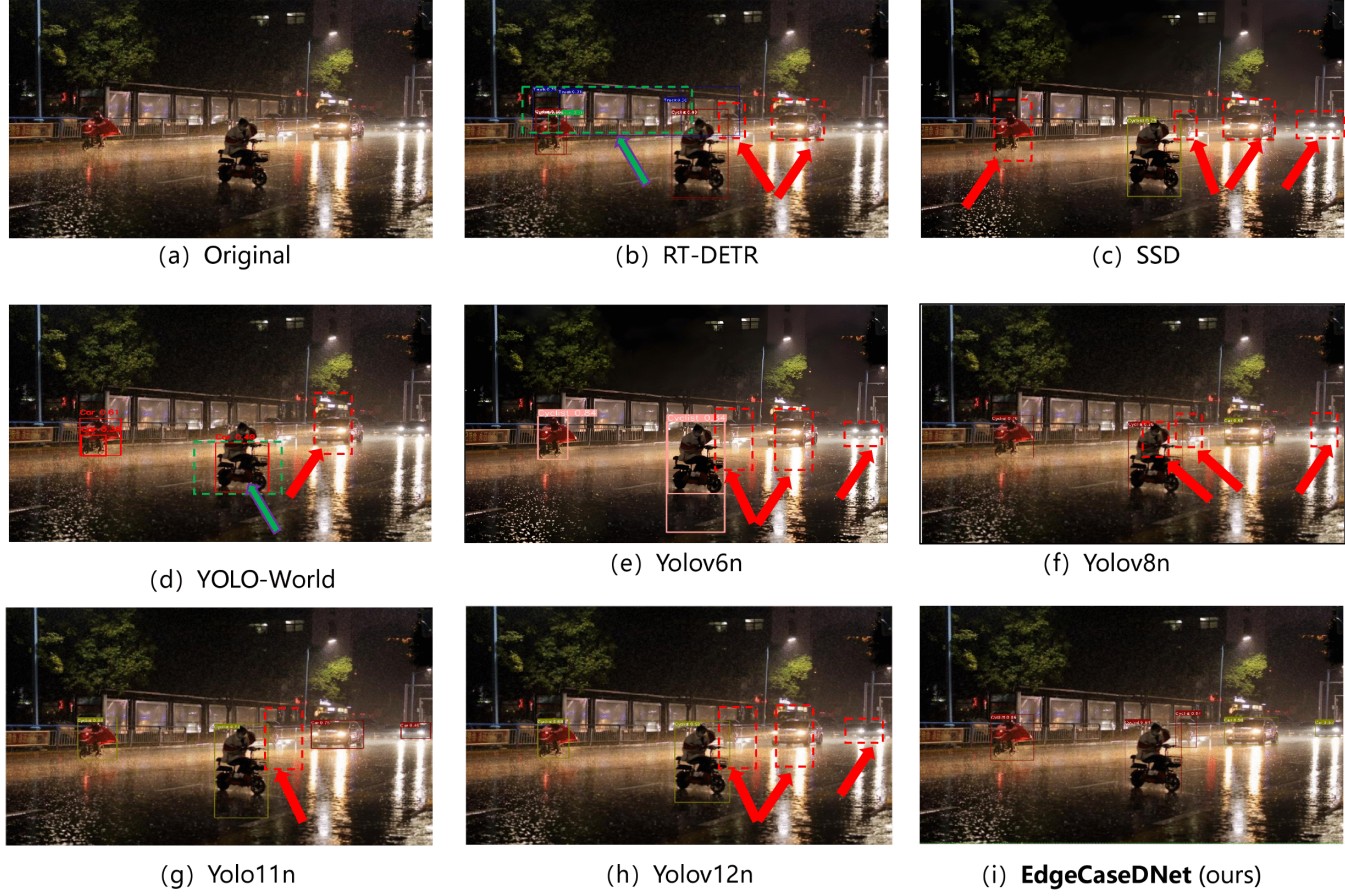

(a) Original (b) RT-DETR (c) SSD

(d) YOLO-World (e) Yolov6n (f) Yolov8n

(g) Yolo11n (h) Yolov12n (i) **EdgeCaseDNet** (ours)

**Fig 8. Comparison of the target detection results of different models in Scenario 3.** The detection results in images (b)–(h) demonstrate that conventional object detection algorithms are more prone to missed and false detections in edge-case scenarios with complex backgrounds, such as rainy nights and the red dashed box indicates the case of target missed detection, while the green dashed box denotes the case of target false detection. In contrast, the EdgeCaseDNet algorithm exhibits superior robustness in these challenging conditions.

## 4.5 Ablation study

To assess the efficacy of the proposed enhancements, we conducted ablation experiments using consistent hyperparameters and training protocols. Table 5 presents the results. Substitution of the original YOLOv8 backbone with HGNetv2 resulted in a 7.0% increase in mAP@50 and a 5.6% increase in mAP@[.5:.95], thereby improving multiscale feature extraction for road targets while reducing parameters and computational demands. The replacement of the original convolutional blocks with the lightweight HPDSConv module reduced the model's GFLOPs by 0.1. The implementation of the Wise-IoU loss function enhances both the mean average precision and the convergence speed of the model (see Figs 11 and 12). Furthermore, the Haar_HGNetv2 module, augmented by the wavelet transform, reduces the computational cost without compromising the performance. The wavelet transform effectively preserves high-frequency information, thereby mitigating edge and detail blurring issues associated with traditional convolutional downsampling. This enhancement facilitates improved detection of small objects and targets in complex scenes. Lastly, the replacement of the baseline detection head with AFPN increased the number of parameters and computations but further improved the performance, with mAP@50 reaching 58.3%.

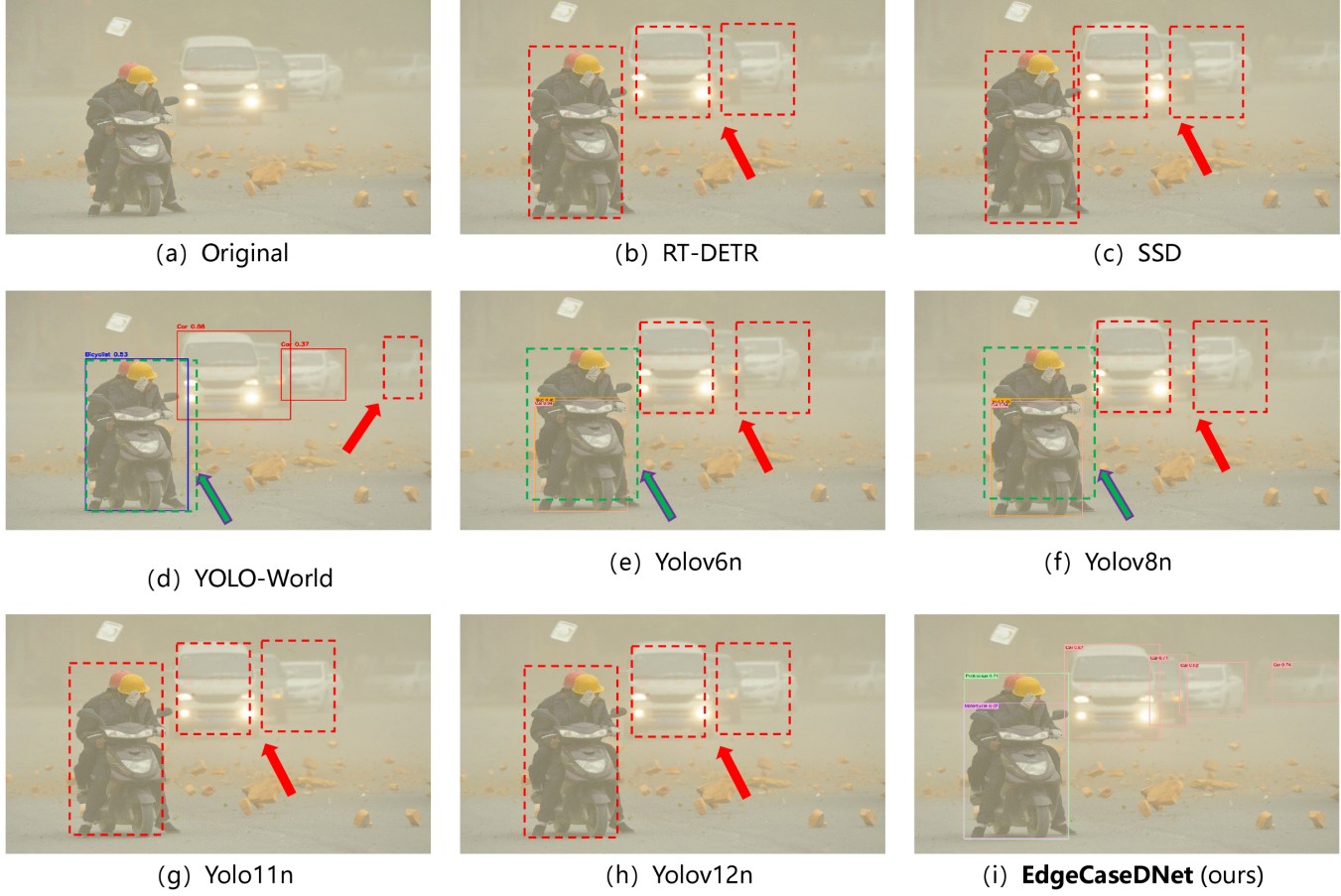

(a) Original

(b) RT-DETR

(c) SSD

(d) YOLO-World

(e) Yolov6n

(f) Yolov8n

(g) Yolo11n

(h) Yolov12n

(i) **EdgeCaseDNet** (ours)

**Fig 9. Comparison of the target detection results of different models in Scenario 4.** The detection results in images (b)–(h) demonstrate that conventional object detection algorithms are more prone to missed and false detections in edge-case scenarios with severe visual degradation. The red dashed box indicates the case of target missed detection, while the green dashed box denotes the case of target false detection. In contrast, the Edge-CaseDNet algorithm exhibits superior robustness in these challenging conditions.

Figs 11 and 12. demonstrated the improvement effects of the YOLOv8 detection algorithm after replacing the loss function with Wise-IoU.

As illustrated in Fig 11, the orange curve (YOLOv8 + HGNetv2 + Wise-IoU) exhibits superior values post-convergence. This observation suggests that the integration of the Wise-IoU loss function into the YOLOv8 detection algorithm enhances both detection precision and recall rate. Furthermore, the orange curve demonstrates marginally better performance in both the IoU = 0.5 and IoU = 0.5–0.95 scenarios.

Fig 12 illustrates the bounding box loss, classification loss, and distribution focal loss observed during the training and validation phases, respectively. A comparison between the orange and blue curves reveals that the orange curve demonstrates superior performance in terms of both the rate of descent and the final stable value. This outcome suggests that the Wise-IoU module contributes positively to the optimization of bounding box regression and classification tasks. The Wise-IoU loss function appears to facilitate a more rapid convergence to the optimal solution through a more effective loss calculation method, while also mitigating the risk of overfitting. The comparison of validation losses further corroborates this conclusion.

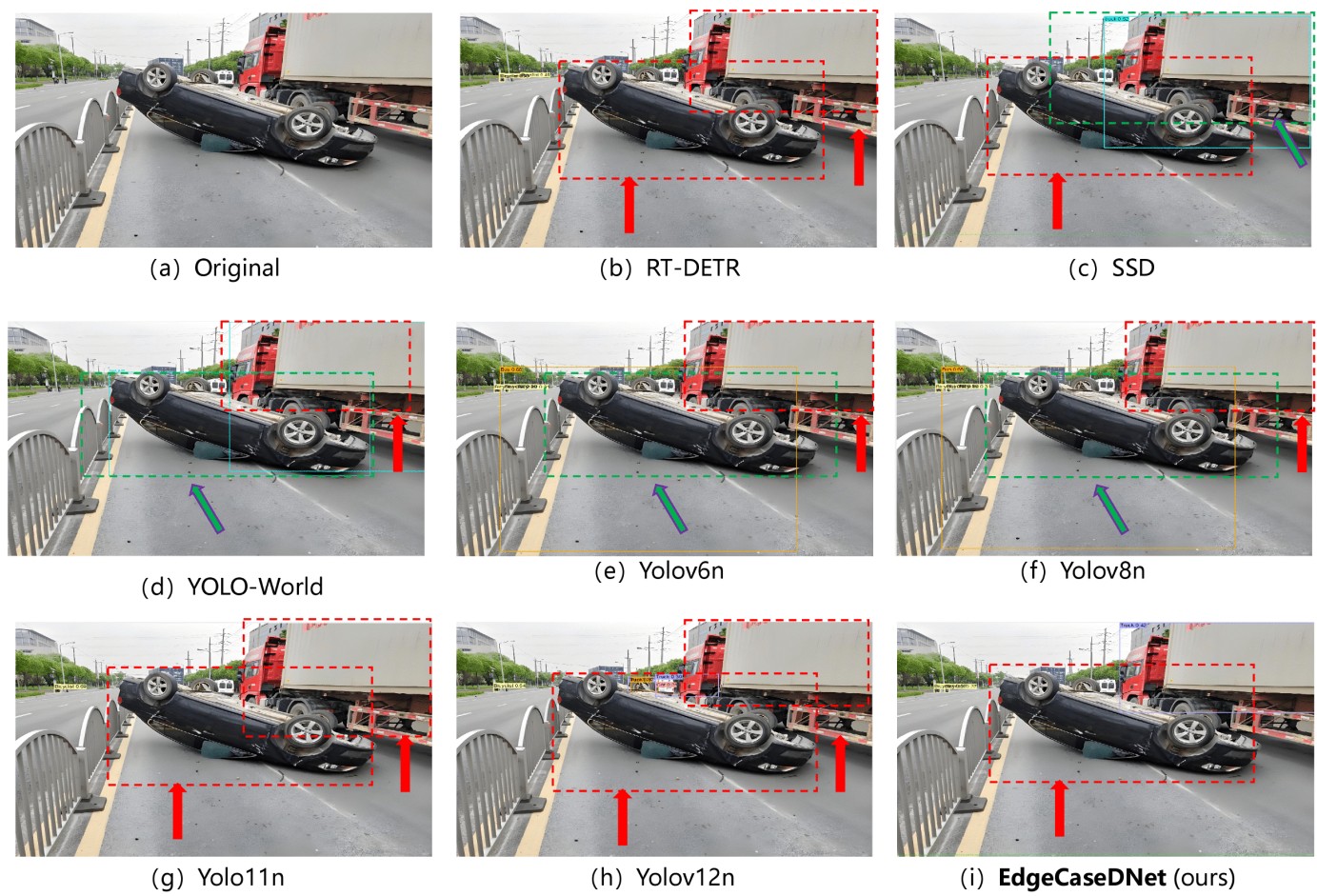

**Fig 10. Comparison of the target detection results of different models in Scenario 5.** The red dashed box indicates the case of target missed detection, while the green dashed box denotes the case of target false detection.

**Table 5. Ablation experiment.**

| Models | | | | | | | Performance | | | |
|---|---|---|---|---|---|---|---|---|---|---|
| Group | Yolov8n | Wise-IoU | Haar_HGNetv2 | HGnetv2 | HPDSConv | AFPN | $AP_{50}^{val}$ | $AP_{50:95}^{val}$ | Params | GFLOPs |
| 1 | ✓ | | | | | | 47.7% | 30.7% | 4.2M | 8.9 |
| 2 | ✓ | ✓ | | | | | 48.1% | 30.7% | 4.2M | 8.9 |
| 3 | ✓ | | | ✓ | | | 54.7% | 36.3% | 2.3M | 6.7 |
| 4 | ✓ | ✓ | | ✓ | | | 54.9% | 36.5% | 2.3M | 6.7 |
| 5 | ✓ | ✓ | | ✓ | ✓ | | 54.5% | 35.6% | 2.3M | 6.6 |
| 6 | ✓ | ✓ | ✓ | | ✓ | | 55.1% | 35.6% | 2.3M | 6.9 |
| 7 | ✓ | ✓ | ✓ | | ✓ | ✓ | **58.3%** | **39.1%** | **2.5M** | **7.0** |

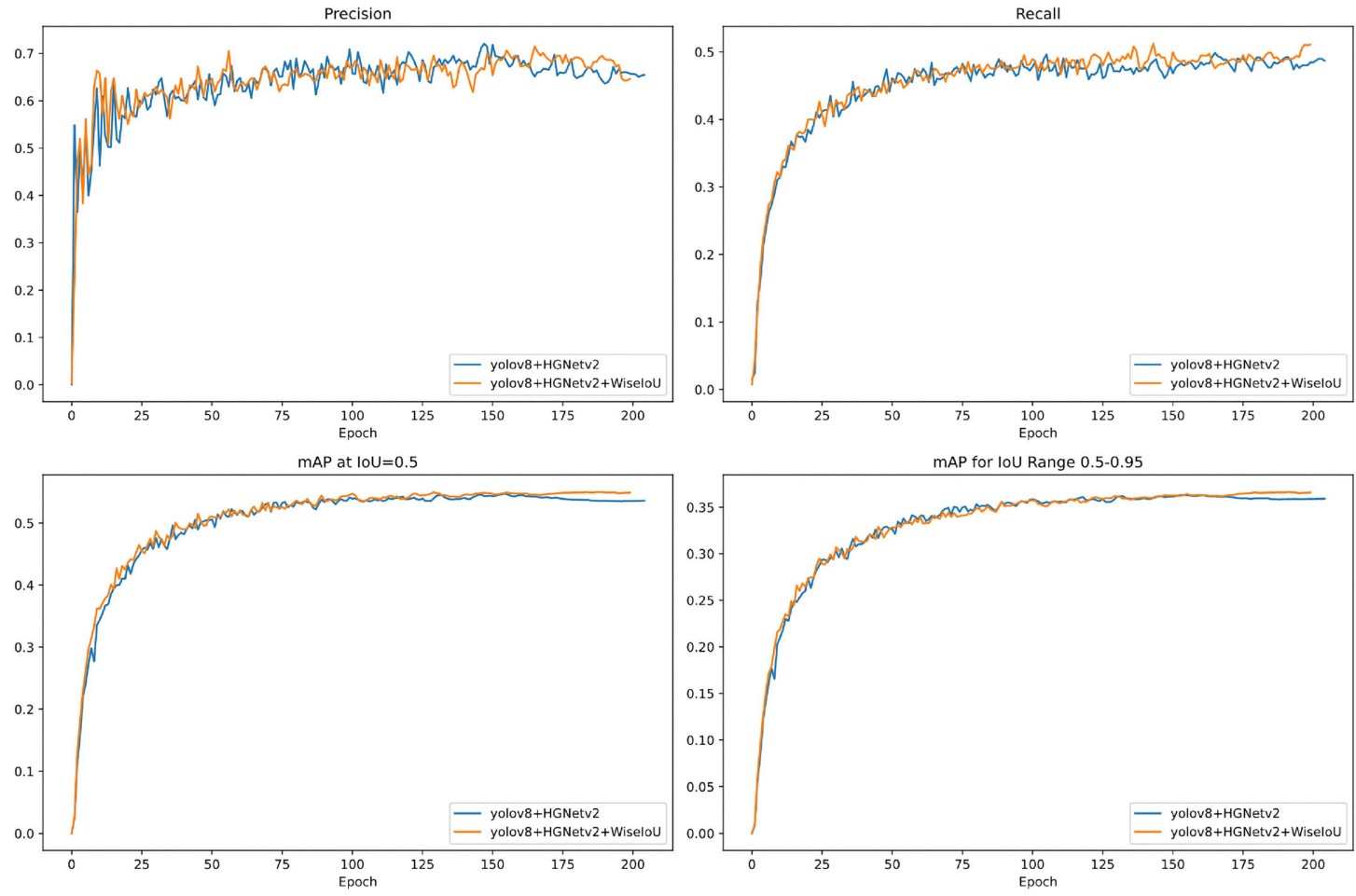

**Fig 11. Performance comparison of the improved YOLOv8 algorithm.** Analysis of the performance of YOLOv8 + HGNetv2 and YOLOv8 + HGNetv2 + Wise-IoU under different metrics.

Fig 13 provides a comparison of the feature maps generated by the conventional HGNetv2 module and Haar_HGNetv2 module.

Collectively, these enhancements simultaneously improved detection accuracy and reduced model size, achieving model lightweighting.

## 5. Conclusions

To address the challenges of low accuracy, high miss rates, and inaccurate object localization in edge-case scenarios for autonomous driving, this study presents a new object detection dataset for edge cases in autonomous driving, DEAP, and proposes an improved object detection framework, EdgeCaseDNet. DEAP, a benchmark dataset covering diverse extreme weather conditions, provides a crucial testing platform for developing robust object detection algorithms and is particularly suitable for verifying the generalization capability and failure boundaries of perception systems under complex meteorological conditions.

The core innovation of EdgeCaseDNet was the replacement of the original backbone of YOLOv8 with an improved CNN-based backbone, Haar_HGNetv2, thereby significantly enhancing the detection accuracy for road targets. This

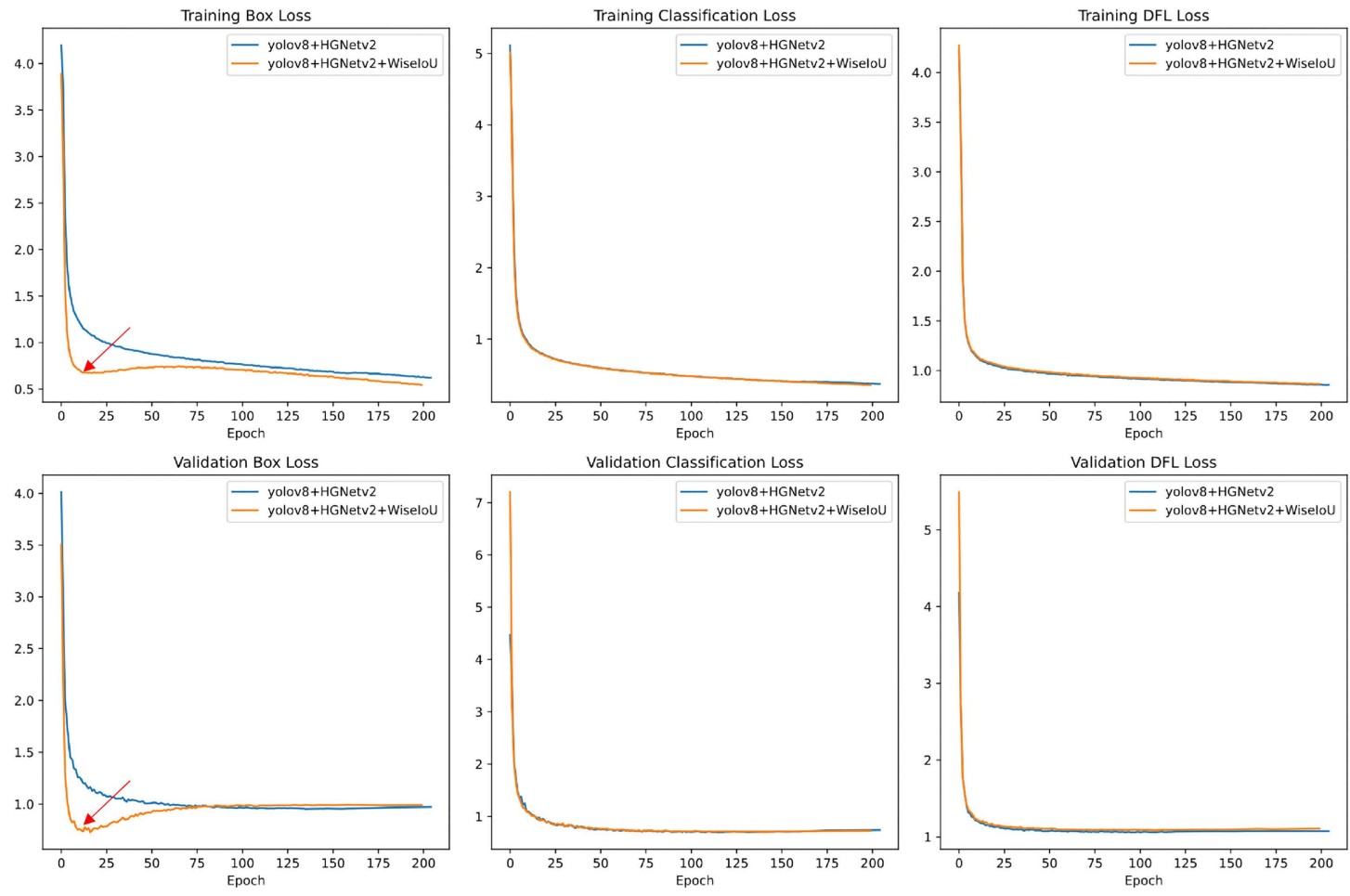

**Fig 12. Comparison of loss curves for the improved YOLOv8 algorithm.** Analysis of training and validation losses between YOLOv8 + HGNetv2 and YOLOv8 + HGNetv2 + Wise-IoU.

modification enables the capture of long-range dependencies between objects, reduces false positives and negatives, and adapts to targets with varying scales and shapes. Additionally, the original feature pyramid network in YOLOv8 was replaced with the AFPN module to further improve the detection performance. The proposed HPDSConv convolution module replaces the original convolution module and reduces network parameters while maintaining accuracy. Moreover, the introduction of the Wise-IoU loss function instead of the original bounding box regression loss function accelerates algorithm convergence and enhances object localization accuracy. Experiments conducted on the DEAP dataset demonstrated that the improved YOLOv8 algorithm maintained high detection speeds while significantly enhancing accuracy, effectively addressing detection performance issues and inaccuracies caused by suboptimal background conditions.

## Author contributions

**Conceptualization:** Wancheng Ge.

**Data curation:** Chen Shan.

**Formal analysis:** Wancheng Ge.

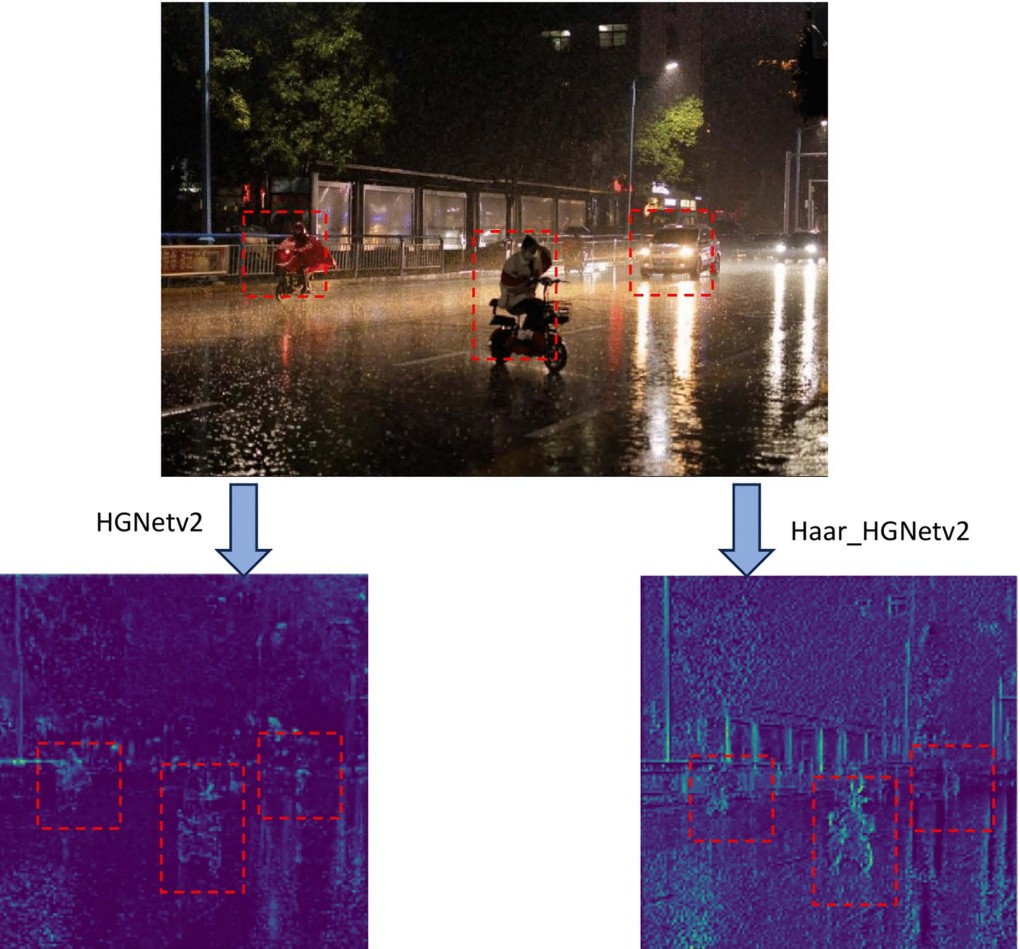

**Fig 13. The comparison of feature maps generated by the conventional HGNetv2 and Haar_HGNetv2 module.** Wavelet transform more effectively preserves high-frequency information, thereby retaining more edge details.

**Funding acquisition:** Wancheng Ge.

**Investigation:** Wancheng Ge.

**Methodology:** Yin Lei, Chen Shan.

**Project administration:** Yin Lei.

**Resources:** Wancheng Ge.

**Software:** Yin Lei, Chen Shan.

**Supervision:** Wancheng Ge.

**Validation:** Yin Lei, Chen Shan.

**Visualization:** Yin Lei, Chen Shan.

**Writing – original draft:** Yin Lei, Chen Shan.

**Writing – review & editing:** Yin Lei.

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
