## [Decision Letter · Decision Letter 0]

28 Apr 2025

Dear Dr. Lei,

Thank you for submitting your manuscript to PLOS ONE. After careful consideration, we feel that it has merit but does not fully meet PLOS ONE’s publication criteria as it currently stands. Therefore, we invite you to submit a revised version of the manuscript that addresses the points raised during the review process.

We look forward to receiving your revised manuscript.

Kind regards,

Alessio Plebe

Academic Editor

PLOS ONE

“This research was supported in part by the International Strategic Innovative Project of the National Key Research and Development Program of China under Grant(No. 2023YFE0112500)”

“This research was supported in part by the International Strategic Innovative Project of the National Key Research and Development Program of China under Grant(No. 2023YFE0112500)”

“This research was supported in part by the International Strategic Innovative Project of the National Key Research and Development Program of China under Grant(No. 2023YFE0112500)”

Additional Editor Comments:

As you can discern from the reviews, significant improvements are needed for the manuscript to reach a publishable standard.

Additionally, it is worth noting that since the manuscript proposes a new software, it must be open source and stored in an appropriate archive, conform to the Open Source Definition, and complete with all necessary information for its installation and execution.

Reviewers' comments:

Reviewer's Responses to Questions

**Comments to the Author**

1. Is the manuscript technically sound, and do the data support the conclusions?

Reviewer #1: No

Reviewer #2: Yes

2. Has the statistical analysis been performed appropriately and rigorously?

Reviewer #1: Yes

Reviewer #2: Yes

3. Have the authors made all data underlying the findings in their manuscript fully available?

Reviewer #1: No

Reviewer #2: Yes

4. Is the manuscript presented in an intelligible fashion and written in standard English?

Reviewer #1: No

Reviewer #2: Yes

Reviewer #1: While the manuscript proposes a promising object detection framework and presents encouraging experimental results, the theoretical foundation contains a critical misinterpretation.

1. Theoretical clarity on HGNetv2 vs RT-DETR. The manuscript incorrectly refers to HGNetv2 as the "RT-DETR backbone network" (e.g., Lines 50–51, 125–126). This is misleading—HGNetv2 is a CNN backbone that can be used within models like RT-DETR, but it is not part of RT-DETR’s transformer-based architecture. This conceptual confusion weakens the theoretical grounding and should be corrected throughout the manuscript to accurately reflect HGNetv2’s role as a standalone CNN backbone integrated with YOLOv8 in this work.

2. Figure 1 notation and clarification. The manuscript refers to “dot lines” in Figure 1, but the lines are red dashed lines. The terminology should be corrected for clarity. Additionally, multiple red dashed boxes are used—each box’s function should be clearly labeled or described in the caption.

3. Figure 2 – incomplete HG-block architecture. Figure 2, titled "The Network Architecture of HG-Block," fails to show the architecture in sufficient detail. Modules such as “CBA[4]”, “CBA[5]”, etc., appear without any explanation. Please define what ConvBAct (CBA) means and describe each block’s function. A clear legend and detailed block-level diagram are recommended.

4. Equation symbols. Several equations throughout the manuscript use variables (e.g., , , , , , etc.) without proper definitions or descriptions. All symbols should be explicitly defined when first introduced to ensure clarity and reproducibility.

5. Figure 4 – Visual overlap issue. In Figure 4, the overlay of text and image elements reduces legibility. Please adjust the layout to ensure all components (e.g., arrows, letters, and labels) are clearly visible and non-overlapping.

6. Check the usage of the language.

Reviewer #2: The manuscript "EdgeCaseDNet: An Enhanced Detection Architecture for Edge Case Perception in Autonomous Driving" addresses critical challenges in the field and proposes several meaningful enhancements to the YOLOv8 architecture. While the experimental results are convincing, showing substantial improvements over the YOLOv8 baseline and other comparison methods, there are several issues with the presentation that should be addressed before publication.

1. While there is an ablation study, it could benefit from a more detailed analysis of why each component contributes to the observed improvements.

2. The construction of the DEAP dataset would strengthen the paper, particularly regarding how edge cases were identified and selected.

3. While the authors mention reducing parameters and computational complexity throughout the paper, it lacks an in-depth computational analysis. A more comprehensive analysis of inference time and computational requirements would strengthen the practical applicability claims.

4. The manuscript contains numerous grammatical errors and awkward phrasings that impede comprehension. For example:

4.1 Line 74-75: "These methods are cumbersome, have poor generalization and robustness, and cannot meet practical road-scenario detection requirements." This is an overgeneralization without supporting evidence.

4.2 Line 149-151: "The Tiny model contains only three LDS Layers to maximize the GPU computation efficiency." The causal relationship is not clearly explained.

4.3 Line 238-239: " ∈[1, e) increases the focus on anchors of average quality." The meaning of "anchors of average quality" is not defined.

5. The term "edge case" is central to the paper, but it is never precisely defined. What constitutes an edge case versus a normal case should be clearly stated.

6. In Section 3.2, "Partial Convolution (PConv)" is introduced, but its mathematical formulation is missing, unlike that of other components.

7. The paper alternates between "HPDSCM" and "HPDSConv" when referring to the same module.

Minor changes:

1. The description of performance improvements appears in nearly identical form in the abstract, introduction, and conclusion.

2. The explanation of the CIoU loss function's limitations is repeated in similar terms in lines 229-232 and again when introducing the Wise-IoU loss.

3. The references to the figures are inconsistent in the manuscript. "Figure 1 shows the...." should be "Fig. 1. shows the...."

4. Figure 1 is cluttered, especially the HPDSCM, which is quite small and difficult to follow.

**Do you want your identity to be public for this peer review?** For information about this choice, including consent withdrawal, please see our Privacy Policy

Reviewer #1: No

Reviewer #2: No

---

## [Author Response · Author response to Decision Letter 1]

25 May 2025

To: PLOS ONE Editor

Re: Response to reviewers

Dear Dr., Alessio Plebe,

Thank you for your constructive feedback and the opportunity to revise our manuscript "EdgeCaseDNet: An Enhanced Detection Architecture for Edge Case Perception in Autonomous Driving" (ID: PONE-D-25-09823). We have carefully addressed all comments from the reviewers and editors. These comments are all valuable and very helpful for revising and improving our paper, as well as the important guiding significance to our research. Based on these comments and suggestions, we have made appropriate modifications to the original manuscript, then we hope to meet with your approval.

According to your and the reviewers’ comments we have addressed four key areas of improvement:

1. Implementation of Decision Letter Guidance

-We have meticulously supplemented and revised the manuscript content in strict accordance with the requirements outlined in your decision letter, ensuring all feedback has been comprehensively addressed.

2. Strengthened Theoretical Rigor

- We have conducted a comprehensive review of the theoretical framework to enhance its clarity, logical soundness, and alignment with established literature.

- We have reinforced the justification for the assumptions and models through expanded logical derivations and comparative analyses.

3. Enhanced Experimental Robustness

- We have expanded the experimental scope.

- We have restructured the Methods section to explicitly detail experimental protocols, evaluation metrics (mAP50-95, FLOPs), and statistical validation methods.

- We have uploaded the source code to a public repository to ensure full reproducibility.

4. Refined Academic Presentation

- We have restructured the manuscript’s narrative flow using transitional phrases and signposting to improve coherence.

We entrusted a professional editing agency to proofread and refine the language.

We are uploading (a) our point-by-point response to the comments (below) (response to reviewers, under “Author’s Response Files”), (b) a revised manuscript (track-changes version), and (c) a clean updated manuscript without highlights (“Main Manuscript”).

Finally, we appreciate all reviewers’ warm work earnestly and hope that the correction will meet with approval. We are grateful for the opportunity to improve our work through this rigorous review process and look forward to your further guidance.

Best regards,

Sincerely,

Yin Lei

---

## [Decision Letter · Decision Letter 1]

15 Sep 2025

Dear Dr. Lei,

We look forward to receiving your revised manuscript.

Kind regards,

Alessio Plebe

Academic Editor

PLOS ONE

Journal Requirements:

Additional Editor Comments:

Reviewer #1:

Reviewer #3:

Reviewers' comments:

Reviewer's Responses to Questions

**Comments to the Author**

Reviewer #1: All comments have been addressed

Reviewer #3: (No Response)

2. Is the manuscript technically sound, and do the data support the conclusions?

Reviewer #1: Yes

Reviewer #3: Yes

3. Has the statistical analysis been performed appropriately and rigorously?

Reviewer #1: Yes

Reviewer #3: No

4. Have the authors made all data underlying the findings in their manuscript fully available?

Reviewer #1: Yes

Reviewer #3: Yes

5. Is the manuscript presented in an intelligible fashion and written in standard English?

Reviewer #1: Yes

Reviewer #3: Yes

Reviewer #1: The revised copy has solved all my concerns in the first round of review process. So, I think the current copy is ready to be published.

Reviewer #3: This paper addresses edge-case perception in autonomous driving by introducing the DEAP dataset (13,030 images from SODA10M and CODA, augmented for adverse weather like fog, rain, snow, and night) and proposing EdgeCaseDNet, a YOLOv8-based model with enhancements including a Haar_HGNetv2 backbone for better feature extraction

1. The core contributions are primarily combinations of existing techniques rather than fundamental innovations. Haar wavelets have been used in computer vision for decades, AFPN and Wise-IoU are from prior work, and HPDSConv appears to be a straightforward combination of depthwise separable and partial convolutions. The authors are encouraged to clarify their contributions more clearly.

2. The DEAP dataset claims comprehensiveness with only 13,030 samples, which seems insufficient for robust edge case representation. The selection criteria from SODA10M and CODA are not clearly explained.

3. The 7:2:1 split is standard, but potential class imbalance (e.g., few tricycles) may bias results. Table 1 lists only a single “train/val number” per class (e.g., Car 49,771 vs Tricycle 346), which suggests severe imbalance yet no stratification strategy, per-condition counts (night/rain/fog/snow), or per-split class statistics are reported. Supply train/val/test counts per class and per condition, plus sampling policies to avoid leakage.

4. The evaluation is limited to the authors' own dataset. This needs more work, as well as clarification about the scope and limitation of the dataset.

5. The comparison primarily includes older YOLO versions and lacks comparison with recent state-of-the-art edge case detection methods.

6. The paper provides no statistical significance testing, limited computational analysis, and insufficient hardware specifications. The three test scenarios are too limited for comprehensive edge case evaluation.

7. While Equation (1) correctly presents the Haar transform, several equations (e.g., Equations 2, 3) unnecessarily complicate standard operations. The notation could be more concise and consistent.

8. The paper lacks analysis of failure cases.

9. Missing detailed computational cost analysis despite efficiency claims.

10. Multiple grammatical errors throughout, including inconsistent terminology ("DEAT dataset" vs "DEAP dataset"), awkward sentence constructions, and unclear technical explanations.

11. The paper provides insufficient justification for why Haar wavelets specifically benefit edge case detection.

12. It seems that the paper mixes mAP50-95 with “mAP50-90” and uses non-standard descriptions.

13. Claims of “real-time” are unsubstantiated: no latency/FPS/memory is reported. GFLOPs are listed in ablations, but input resolution is unspecified, making them uninterpretable. Add end-to-end latency (ms), FPS, peak memory, and evaluation thresholds on the same hardware.

**Do you want your identity to be public for this peer review?** For information about this choice, including consent withdrawal, please see our Privacy Policy

Reviewer #1: No

Reviewer #3: No

---

## [Author Response · Author response to Decision Letter 2]

23 Nov 2025

Dear Editor&reviewers

Thank you for your constructive feedback and the opportunity to revise our manuscript "EdgeCaseDNet: An Enhanced Detection Architecture for Edge Case Perception in Autonomous Driving" (ID: PONE-D-25-09823R1). We sincerely appreciate the thoughtful comments and constructive suggestions provided by the reviewers. We have carefully considered each point and have made extensive revisions to the manuscript, which we believe have significantly strengthened it.

We have provided a detailed, point-by-point response to all comments below. All changes in the manuscript have been highlighted for your convenience.

We hope that you find our revisions satisfactory and that the manuscript is now acceptable for publication in PLOS One.

Sincerely,

Yin Lei

E-mail: 001leiyin@tongji.edu.cn

---

## [Editor Report · Decision Letter 2]

26 Nov 2025

EdgeCaseDNet: An enhanced detection architecture for edge case perception in autonomous driving

PONE-D-25-09823R2

Dear Dr. Lei,

We’re pleased to inform you that your manuscript has been judged scientifically suitable for publication and will be formally accepted for publication once it meets all outstanding technical requirements.

Kind regards,

Alessio Plebe

Academic Editor

PLOS ONE

Additional Editor Comments (optional):

This latest version of the manuscript has satisfactorily resolved the remaining

comments that emerged in the last round of review.

Therefore, the work can now be considered ready for publication.
---

## [Editor Report · Acceptance letter]

PONE-D-25-09823R2

PLOS ONE

Dear Dr. Lei,

I'm pleased to inform you that your manuscript has been deemed suitable for publication in PLOS ONE. Congratulations! Your manuscript is now being handed over to our production team.

Kind regards,

on behalf of

Dr. Alessio Plebe

Academic Editor

PLOS ONE